# Effects of Different Types of Carbohydrates on Arterial Stiffness: A Comparison of Isomaltulose and Sucrose

**DOI:** 10.3390/nu13124493

**Published:** 2021-12-15

**Authors:** Ryota Kobayashi, Miki Sakazaki, Yukie Nagai, Kenji Asaki, Takeo Hashiguchi, Hideyuki Negoro

**Affiliations:** 1Center for Fundamental Education, Teikyo University of Science, Tokyo 120-0045, Japan; 2Research & Development Division, Mitsui Sugar Co., Ltd., Tokyo 103-8423, Japan; Miki.Sakazaki@mitsui-sugar.co.jp (M.S.); Yukie.Nagai@mitsui-sugar.co.jp (Y.N.); 3Department of Tokyo Judo Therapy, Teikyo University of Science, Tokyo 120-0045, Japan; k-asaki@ntu.ac.jp; 4Department of School Education, Teikyo University of Science, Tokyo 120-0045, Japan; hasiguti@ntu.ac.jp; 5Harvard PKD Center for Polycystic Kidney Disease Research, Boston, MA 02115, USA; oystercope@gmail.com; 6Faculty of Medicine, Nara Medical University, Nara 634-8521, Japan

**Keywords:** arterial stiffness, glucose ingestion, middle-aged and older patients, isomaltulose, sucrose

## Abstract

Increased arterial stiffness during acute hyperglycemia is a risk factor for cardiovascular disease, but the type of carbohydrate that inhibits it is unknown. The purpose of this study was to determine the efficacy of low-glycemic-index isomaltulose on arterial stiffness during hyperglycemia in middle-aged and older adults. Ten healthy middle-aged and older adult subjects orally ingested a solution containing 25 g of isomaltulose (ISI trial) and sucrose (SSI trial) in a crossover study. In the SSI trial, the brachial–ankle (ba) pulse wave velocity (PWV) increased 30, 60, and 90 min after ingestion compared with that before ingestion (*p* < 0.01); however, in the ISI trial, the baPWV did not change after ingestion compared with that before ingestion. Blood glucose levels 30 min after intake were lower in the ISI trial than in the SSI trial (*p* < 0.01). The baPWV and systolic blood pressure were positively correlated 90 min after isomaltulose and sucrose ingestion (*r* = 0.640, *p* < 0.05). These results indicate that isomaltulose intake inhibits an acute increase in arterial stiffness. The results of the present study may have significant clinical implications on the implementation of dietary programs for middle-aged and elderly patients.

## 1. Introduction

Previous studies have reported acute hyperglycemia as an independent risk factor for cardiovascular disease [1]. An increased postprandial blood glucose level is also a risk factor for cardiovascular disease and exerts a greater effect than the fasting blood glucose level [2]. Moreover, increased arterial stiffness owing to impaired vascular endothelial function underlies the increased risk of cardiovascular disease in acute hyperglycemia [3]. Gordin et al. [4] suggested that arterial stiffness increased with increasing postprandial blood glucose levels in healthy middle-aged and older individuals. Furthermore, we previously demonstrated that systemic arterial stiffness increased in middle-aged and older people after the ingestion of a 25 g glucose solution [5]. Since postprandial blood glucose increases with age [6] and arterial stiffness progresses, there is a significant relationship between arterial stiffness and postprandial blood glucose levels [7]. Japan has a super-aging society [8]; it is important to control the progression of arterial stiffness during acute hyperglycemia in older Japanese individuals.

The increase in arterial stiffness occurs immediately after food intake [7] and may be influenced by the glycemic index (GI) value of the cardiovascular disease indices [9]. In fact, vascular endothelial function, the underlying mechanism of arterial stiffness, varies with GI [10]. A previous study reported that arterial stiffness increased in middle-aged and older people consuming a glucose solution, but the changes differed between different carbohydrate intakes [5]. A high-GI diet has been shown to increase arterial stiffness compared with a low-GI diet. In other words, a low-GI diet may reduce the acute adverse effects on arterial stiffness [11]. For example, isomaltulose is a natural carbohydrate found in honey that has a low GI and is certified as a novel food (European Food Safety Authority) because of its nutritional quality [12]. Isomaltulose has similar amounts of sweetness and energy to sucrose; however, the rate at which it is broken down in the small intestine is slower than that of sucrose, which moderates the rise in blood glucose levels after ingestion [13]. In a previous study, a comparison of acute changes in blood glucose levels after the ingestion of isomaltulose or sucrose in 10 healthy subjects showed that the highest blood glucose levels were lower in patients who ingested isomaltulose than in those who ingested sucrose [14]. In addition, when 10 patients with type 2 diabetes were asked to consume either isomaltulose or sucrose and the changes in blood glucose levels after ingestion were examined, it was found that the blood glucose levels rose rapidly after sucrose ingestion and increased gradually after isomaltulose ingestion, with the peak values being lower for isomaltulose than for sucrose [13]. In other words, isomaltulose, which has a lower GI than sucrose, is expected to reduce the increase in arterial stiffness. However, the changes in arterial stiffness after ingestion of isomaltulose compared with that after the ingestion of sucrose are not sufficiently clear. Therefore, it is necessary to investigate whether arterial stiffness is altered after isomaltulose intake in healthy middle-aged and older people.

In this study, we hypothesized that sucrose intake will increase arterial stiffness with increasing blood glucose levels, but isomaltulose intake will not influence arterial stiffness. To test this hypothesis, we investigated the acute effects of isomaltulose and sucrose intake on arterial stiffness.

## 2. Materials and Methods

### 2.1. Participants

The participants were 10 healthy middle-aged and older adults (five men and five women). Participants were recruited by distributing flyers for research cooperation with residents of the Teikyo University of Science. Finally, we received 20 applications, from which we selected 10 participants who met the following conditions. All participants were normotensive (Japanese standard: <140/90 mmHg), non-smokers, no obvious disease on electrocardiogram or other diagnostic tests, and no exercise habit before the study according to the physical activity questionnaire. Patients with abnormalities in blood tests, urine tests, chest radiographs, or electrocardiograms in the year prior to the study; with diabetes mellitus (American Diabetes Association/ European Association for the Study of Diabetes diagnostic criteria); and who had problems with exercise (e.g., those with musculoskeletal injuries) were excluded from the study. This study was conducted in compliance with the Declaration of Helsinki in terms of ethics, human rights, and protection of participants’ personal information. Ethical approval for this study was obtained from the Ethics Committee of Teikyo University of Science (approval no. 20A013). In addition, this study was registered with the University Hospital Medical Information Network Center (UMIN Center; Study No. UMIN000041622). All hardcopy (paper) study data were stored in a locked filing cabinet, and electronic data were stored on a secured network drive, accessible only to those working in the laboratory. The study was conducted in accordance with the guidelines for human experimentation published by the Institutional Review Board.

### 2.2. Study Design

The participants were 10 healthy middle-aged and older adults. They were instructed to maintain a normal diet and activities of daily living for the duration of the study. Intense exercise (training and activities of daily living), caffeine, and alcohol consumption were prohibited for 24 h prior to the experiment. Fasting (10–12 h) was started at 9:00 p.m. the day before the start of the experiment. Arterial stiffness, blood pressure (BP) at the level of the brachial artery and at the ankle, heart rate (HR), and blood glucose (BG) levels were measured before (baseline) and 30, 60, and 90 min after 25-g isomaltulose or sucrose loading. Before each measurement, the subjects were asked to rest in a supine position (Figure 1).

### 2.3. Body Composition

Height was measured using a height meter in increments of 0.1 cm. Body weight, body fat percentage, and body mass index (BMI) were measured in 0.1 kg increments using a precision instrument body-composition analyzer (WB-150 PMA, Tanita, Tokyo, Japan).

### 2.4. Arterial Stiffness

Pulse wave velocity (PWV) at the brachial and ankle (ba) and at the brachial and heart (hb) of all participants was measured using an automated oscillometric device (PWV/Ankle Brachial Index (ABI), Colin Medical Technology, Komaki, Japan) as previously described [15]. All measurements were performed in a supine position in a quiet room at baseline and 30, 60, and 90 min after isomaltulose solution and sucrose solution ingestion. The daily coefficients of variation in our laboratory were 3 ± 1% and 3 ± 2% for baPWV and hbPWV, respectively.

### 2.5. Upper Arm and Ankle Blood Pressure

Systolic blood pressure (SBP), mean blood pressure (MBP), diastolic blood pressure (DBP), and pulse pressure (PP) of the upper arm and ankle were measured in the supine position using an automated oscillometric PWV/ABI device (Omron Colin, Tokyo, Japan) over the brachial and posterior tibial arteries [15]. All measurements were performed in the supine position in a quiet room at baseline and 30, 60, and 90 min after isomaltulose solution and sucrose solution ingestion. The coefficients of variation per day in our laboratory were 2 ± 1% and 2 ± 2% for brachial blood pressure and ankle blood pressure, respectively.

### 2.6. Heart Rate

HR was measured in the supine position using an automated oscillometric PWV/ABI device (Omron Colin, Tokyo, Japan) [15]. All measurements were performed in the supine position in a quiet room at baseline and 30, 60, and 90 min after the ingestion of isomaltulose and sucrose solutions. The coefficient of variation per day in our laboratory was 2 ± 1%.

### 2.7. Blood Glucose

Venous blood was collected from the participants’ left fingertips. Blood glucose levels were measured by the flavin-adenine dinucleotide glucose dehydrogenase method using a Glutest Neo Alpha glucometer (Sanwa Kagaku Kenkyusho, Tokyo, Japan) [16]. Measurements were taken before and 30, 60, and 90 min after ingestion of isomaltulose and sucrose solutions. The interday coefficient of variation of blood glucose levels was 3 ± 1%.

### 2.8. Isomaltulose Solution and Sucrose Solution Ingestion

Each participant orally ingested 25 g of isomaltulose (ISI trial) or 25 g of sucrose (SSI trial) in 200 mL of water within 5 min, since the new World Health Organization guidelines recommend that adults consume less than 25 g of free sugars per day [17]. Each subject waited approximately 3 days after the completion of one test before taking the next test.

### 2.9. Statistical Analysis

Data are presented as means ± standard deviation. Normality of the data and homogeneity of variance were examined using the Shapiro–Wilk and Levene tests, respectively. Changes in each measurement before and after the intervention are presented as mean values and 95% confidence intervals for each group. Parametric analysis was performed using two-way analysis of variance with repeated measures (time*group) for the measurements taken. When the assumption of sphericity was violated (Mauchly’s test), the analysis was adjusted using the Greenhouse–Geisser correction. The Bonferroni method was used with post hoc tests for changes in each intervention. The total area under the curve at 90 min (AUC) was calculated using the trapezoidal formula and analyzed using the corresponding *t*-test. The correlation between baPWV and brachial SBP levels 90 min after consumption was examined using the Pearson product-moment correlation coefficient. SPSS (version 25, IBM Corp., Armonk, NY, USA) was used for the statistical analysis. Statistical significance was set at α = 0.05, and all α values were two-sided. To examine the magnitude of the differences, the effect size was calculated based on Cohen’s d.

## 3. Results

### 3.1. Physical Characteristics

The mean age of the participants was 62.8 ± 4.4 years; the mean height was 162.5 ± 2.9 cm; the mean weight was 60.9 ± 3.0 kg; the mean BMI was 23.1 ± 1.1 kg/m^2^; and the mean body fat percentage was 27.7 ± 2.7% (Table 1).

### 3.2. Arterial Stiffness

In the SSI trial, the baPWV increased 30, 60, and 90 min after ingestion compared with that before ingestion (*p* < 0.01); however, in the ISI trial, the baPWV did not change after ingestion compared with that before ingestion. The baPWV was not significantly different between the trials before ingestion (Figure 2A). The baPWV AUC was lower (*p* < 0.01) in the ISI trial than in the SSI trial (Figure 2B).

The hbPWV did not change after ingestion compared with that before ingestion in both the SSI and ISI trials, and the hbPWV was not different between the two trials (Figure 2C). The hbPWV AUC did not differ between the trials (Figure 2D).

### 3.3. Heart Rate

The HR did not change after sucrose ingestion compared with that before sucrose ingestion. Moreover, the HR did not change after isomaltulose ingestion compared with that before isomaltulose ingestion. Furthermore, there was no difference between the trials (Table 2).

### 3.4. Brachial Blood Pressure

The SBP and PP of the upper arm in the SSI trial increased 90 min after ingestion compared with those before ingestion (*p* < 0.05), whereas the SBP and PP of the upper arm in the ISI trial did not change after ingestion compared with those before ingestion. There was no difference between the trials. The MBP and DBP of the upper arm in the SSI trial did not change after ingestion compared with those before ingestion. The MBP and DBP of the upper arm in the ISI trial did not change after ingestion compared with those before ingestion. There was no difference between trials (Table 2).

### 3.5. Ankle Blood Pressure

The SBP, MBP, and PP of the ankle in the SSI trial increased 90 min after ingestion compared with those before ingestion (*p* < 0.05), and the SBP, MBP, and PP of the ankle in the ISI trial did not change after ingestion compared with those before ingestion. There were no differences between the trials. 

The DBP of the ankle in the SSI trial did not change after ingestion compared with that before ingestion, and the DBP and HR of the ankle in the ISI trial did not change after ingestion compared with those before ingestion. There were no differences between the trials (Table 3). 

### 3.6. Blood Glucose

The blood glucose levels in the SSI trial increased 30 and 60 min after ingestion compared with those before ingestion (*p* < 0.01). The blood glucose levels in the ISI trial increased 30 min after ingestion compared with those before ingestion (*p* < 0.05). The blood glucose levels 30 min after intake were lower in the ISI trial than in the SSI trial (*p* < 0.01, (Figure 3A). The AUC of blood glucose level was lower in the ISI trial than in the SSI trial (Figure 3B).

### 3.7. Arterial Stiffness and Brachial SBP at 90 Min after Sucrose and Isomaltulose Solution Ingestion

The baPWV and brachial SBP were positively correlated 90 min after isomaltulose and sucrose ingestion (*r* = 0.640, *p* < 0.05) (Figure 4).

## 4. Discussion

The main finding of this study was that the baPWV and SBP did not change after isomaltulose intake compared to before. This confirms our hypothesis. These results suggest that isomaltulose could be used as an alternative to sucrose, given the neutral effect on the PWV and SBP.

A rapid increase in blood glucose levels after a meal is an independent risk factor for cardiovascular disease and a greater risk factor than fasting glucose [18]. Therefore, it is necessary to control the rapid increase in blood glucose levels after meals to prevent cardiovascular diseases. There is a consensus that eating high-GI foods results in rapid carbohydrate absorption, whereas low-GI foods result in milder carbohydrate absorption and consequently milder insulin secretion [19]. For example, a previous study of 10 healthy individuals showed a slower increase in blood glucose and insulin levels after consuming isomaltulose compared with that after sucrose consumption [14]. Our results are in agreement with these findings. Blood glucose levels after ingestion were lower in the ISI trial than in the SSI trial. Therefore, isomaltulose may slow down the rise in blood glucose levels after a meal compared with sucrose, the main component of sugar.

A number of studies have shown that arterial stiffness increases during hyperglycemia [4,19,20]. Moreover, previous studies have shown that the baPWV increases during acute hyperglycemia [5]. Our previous study also found an increase in the baPWV after glucose ingestion. The present results are in agreement with these findings, in which the baPWV increased 30, 60, and 90 min after sucrose ingestion compared with before sucrose ingestion, but no increase was observed in the ISI trial. In addition, the AUC of the baPWV was lower in the ISI trial than in the SSI trial. Therefore, isomaltulose can be expected to inhibit the increase in arterial stiffness compared with the consumption of other carbohydrates, such as sucrose and glucose, making it possible to create food products that are both tasty and healthy.

Diabetes mellitus induces peripheral arterial disease in the limbs [21] and it has been found that peripheral arterial stiffness increases after the 75-g glucose-tolerance test compared with before the test [22]. Previous studies have reported that the peripheral arterial PWV, especially in the lower limb arteries, increases during acute hyperglycemia [23]. In the current SSI study, the baPWV increased after ingestion compared to before sucrose ingestion, while the hbPWV did not change. Previous studies have reported that the baPWV reflects arterial stiffness in the distal (mainly abdominal) aorta and lower limbs [24], while the hbPWV reflects arterial stiffness in the proximal aorta and upper limbs [25]. In addition, the MBP and PP, which reflect aortic and peripheral arterial stiffness, are elevated after sucrose ingestion. Therefore, in middle-aged and older people, an increase in arterial stiffness during acute hyperglycemia is likely to affect the abdominal aorta and lower-limb arteries. However, in the current study, we were unable to measure arterial stiffness in detail by site. In future studies, we plan to further investigate the increase in arterial stiffness during acute hyperglycemia by site.

This study did not examine the mechanism by which arterial stiffness was not altered after isomaltulose ingestion, but there are several possible explanations. In the present study, in the SSI test, the SBP increased 90 min after compared to before sucrose intake. In previous studies, the SBP and baPWV were found to be correlated [26]. In the present study, there was a correlation between the SBP and baPWV in the upper arm after 90 min of ingestion (*r* = 0.640, *p* < 0.05). This suggests that increased systemic arterial stiffness may be responsible for the increase in the SBP. In this study, there was no correlation between the baPWV and the blood glucose level at 90 min, when the increase in the baPWV was highest. In other words, the blood glucose level may not be directly involved in the increase in the baPWV during acute hyperglycemia. Furthermore, sympathetic hyperactivity, increased oxidative stress, and decreased vascular endothelial function associated with increased blood glucose levels may be related in parallel. Increased sympathetic nerve activity has been implicated in the increase in the baPWV [27]. Sympathetic nerve activity has been found to increase after eating [28]. The sympathetic ratio after a meal shows a sustained elevation lasting at least one hour, which has been suggested to be primarily due to a decrease in vagal activity [28]. Thus, the increase in the baPWV after sucrose consumption in the present study may be due to increased sympathetic nerve activity. However, since sympathetic nerve activity was not measured in this study, it should be assessed in the future. Decreased vascular function (PWV and FMD) after a meal has been proven to be dependent on oxidative stress [3]. For example, 2-thiobarbituric-acid-reactive substances (TBARS), an indicator of oxidative stress, have been shown to increase after acute hyperglycemia [29]. Oxidative stress is thought to reduce vascular function by increasing asymmetric dimethylarginine (ADMA) [30]. Hyperglycemia-induced vascular dysfunction after an oral glucose challenge was found to be associated with increased plasma ADMA/Arg [30]. Therefore, it is likely that acute hyperglycemia increased TBARS and induced vascular endothelial dysfunction via increased ADMA, which caused the increase in the PWV. However, oxidative stress was not measured in this study and should be measured in future studies.

Regarding the application of the study results, the use of isomaltulose as a sweetener in everyday cooking may reduce arterial stiffness and blood pressure increases during acute hyperglycemia compared with the use of other carbohydrates. We believe that the need for isomaltulose to prevent arteriosclerosis and elevated blood pressure will increase, especially as people are increasingly cooking for themselves as a way of preventing new coronavirus infections and as they become more conscious of nutritional balance.

Nevertheless, this study has certain limitations. One limitation was the relatively small number of participants. However, the sample size was statistically significant. Furthermore, we believe that the findings are not generalizable to different populations (e.g., young people and people with diabetes) because the study was conducted in older people, and we will therefore examine different groups of people in the future. In addition, although insulin and endothelial dysfunction may alter the PWV, insulin levels and endothelial function biomarkers were not measured in the present study.

## 5. Conclusions

The main finding of this study was that the baPWV and SBP did not change after isomaltulose intake compared to before. This confirms our hypothesis. These results suggest that isomaltulose could be used as an alternative to sucrose, given the neutral effect on the PWV and SBP.

## Figures and Tables

**Figure 1 nutrients-13-04493-f001:**
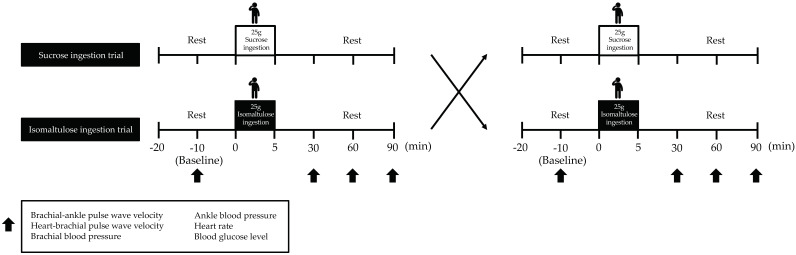
Study design. Arterial stiffness, BP, HR, and BG levels were measured at baseline and at 30, 60, and 90 min after isomaltulose or sucrose ingestion. The participants rested in a supine position for 10 min before the test. BP, blood pressure; HR, heart rate; BG, blood glucose.

**Figure 2 nutrients-13-04493-f002:**
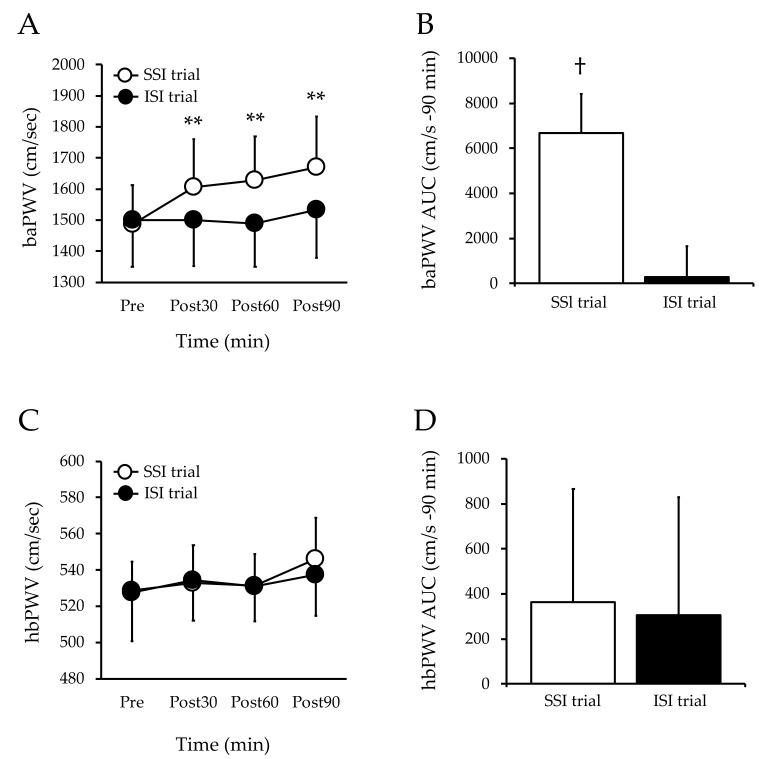
Changes in arterial stiffness at baseline and post-ingestion in both trials. Values are mean ± SD. ** *p* < 0.01 vs. baseline. † *p* < 0.01 vs. ISI trial. baPWV, brachial–ankle pulse wave velocity; hbPWV, heart–brachial pulse wave velocity; SSI, sucrose solution intake; ISI, isomaltulose solution intake; AUC, area under the curve; SD, standard deviation; Figure A, baPWV; Figure B, baPWV AUC; Figure C, hbPWV; Figure D, hbPWV AUC.

**Figure 3 nutrients-13-04493-f003:**
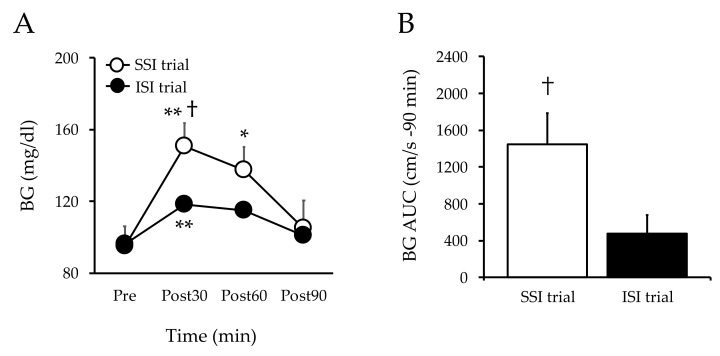
Changes in blood glucose at baseline and post-ingestion in both trials. Values are mean ± SD. ** *p* < 0.01 and * *p* < 0.05, vs. baseline. ^†^
*p* < 0.05, vs. ISI. BG, blood glucose; SSI, sucrose solution intake; ISI, isomaltulose solution intake; SD, standard deviation; Figure 3A, blood glucose; Figure 3B, blood glucose AUC.

**Figure 4 nutrients-13-04493-f004:**
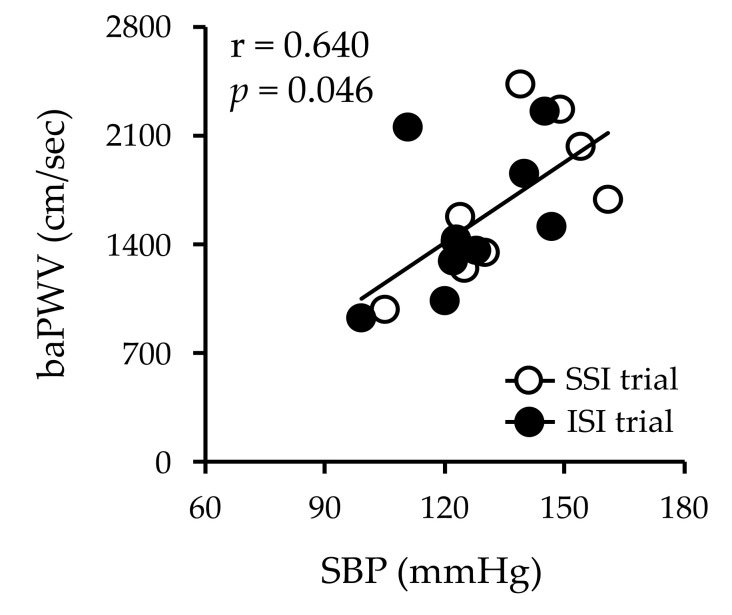
Correlation between arterial stiffness and brachial SBP at 90 min after sucrose and isomaltulose intake. Values are mean ± SD. *r* = 0.640 and *p* = 0.046. SBP, systolic blood pressure; SSI, sucrose solution intake; ISI, isomaltulose solution intake; SD, standard deviation.

**Table 1 nutrients-13-04493-t001:** Baseline characteristics of the participants.

	Value
Age, years	62.8 ± 4.4
Height, cm	162.5 ± 2.9
Weight, kg	60.9 ± 3.0
BMI, kg/m^2^	23.1 ± 1.1
Body fat, %	27.7 ± 2.7
Brachial SBP, mmHg	123.5 ± 6.2
Ankle SBP, mmHg	153.8 ± 8.6
Heart rate, bpm	62.0 ± 3.7
Fasting blood glucose, mg/dL	98.8 ± 4.2

Values are mean ± SD. BMI, body mass index; SBP, systolic blood pressure; SD, standard deviation.

**Table 2 nutrients-13-04493-t002:** Changes in brachial SBP, MBP, DBP, and HR before and after the ingestion of isomaltulose and sucrose.

Variable	Trial	Baseline	Post 30 min	Post 60 min	Post 90 min	*p*-Value (Group)
Brachial SBP, mmHg	SSI trial	123.5 ± 6.2	128.6 ± 6.3	131.0 ± 6.1	134.4 ± 5.9 *	0.93
	ISI trial	124.2 ± 4.6	127.1 ± 4.1	129.4 ± 6.1	126.1 ± 5.3	
Brachial MBP, mmHg	SSI trial	88.2 ± 2.9	90.5 ± 3.2	91.5 ± 2.9	93.4 ± 3.1	0.85
	ISI trial	88.6 ± 2.9	89.8 ± 2.7	91.8 ± 3.8	90.4 ± 3.4	
Brachial DBP, mmHg	SSI trial	70.6 ± 2.5	71.5 ± 2.3	71.7 ± 2.2	72.8 ± 3.0	0.80
	ISI trial	70.8 ± 2.6	71.1 ± 2.6	73.0 ± 3.3	72.6 ± 3.0	
Brachial PP, mmHg	SSI trial	52.9 ± 6.0	57.0 ± 5.3	59.3 ± 5.7	61.6 ± 6.2 *	0.80
	ISI trial	53.4 ± 3.8	56.0 ± 3.7	56.4 ± 4.9	53.6 ± 4.2	
HR, beats/min	SSI trial	62.0 ± 3.7	60.8 ± 2.7	58.1 ± 3.1	58.7 ± 2.5	0.50
	ISI trial	58.1 ± 4.1	54.7 ± 3.2	55.4 ± 2.7	57.4 ± 3.0	

Values are mean ± SD. * *p* < 0.05, vs. baseline. SSI, sucrose solution intake; ISI, isomaltulose solution intake; SBP, systolic blood pressure; MBP, mean blood pressure; DBP, diastolic blood pressure; PP, pulse pressure; HR, heart rate; SD, standard deviation.

**Table 3 nutrients-13-04493-t003:** Changes in ankle SBP, MBP, and DBP before and after the ingestion of isomaltulose and sucrose.

Variable	Trial	Baseline	Post 30 min	Post 60 min	Post 90 min	*p*-Value (Group)
Ankle SBP, mmHg	SSI trial	153.8 ± 8.6	160.6 ± 8.8	165.3 ± 10.2	167.4 ± 8.0 *	0.93
	ISI trial	147.9 ± 9.9	155.1 ± 8.2	154.0 ± 8.1	155.7 ± 9.6	
Ankle MBP, mmHg	SSI trial	99.0 ± 3.1	102.4 ± 3.5	104.1 ± 3.4	106.3 ± 3.2 *	0.85
	ISI trial	95.4 ± 4.7	99.9 ± 3.3	99.8 ± 3.8	100.0 ± 4.4	
Ankle DBP, mmHg	SSI trial	71.6 ± 1.7	73.2 ± 1.7	73.6 ± 2.0	75.8 ± 2.8	0.80
	ISI trial	69.2 ± 3.2	72.2 ± 2.6	72.7 ± 2.9	72.2 ± 3.1	
Ankle PP, mmHg	SSI trial	82.2 ± 8.8	87.4 ± 8.4	91.7 ± 10.7	91.6 ± 8.5 *	0.80
	ISI trial	78.7 ± 8.9	82.9 ± 8.5	81.3 ± 7.6	83.4 ± 9.1	

Values are mean ± SD. * *p* < 0.05, vs. baseline. SSI, sucrose solution intake; ISI, isomaltulose solution intake; SBP, systolic blood pressure; MBP, mean blood pressure; DBP, diastolic blood pressure; PP, pulse pressure; SD, standard deviation.

## Data Availability

The data presented in this study are available from the corresponding author upon request.

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
