# Peer review of "Effects of Different Types of Carbohydrates on Arterial Stiffness: A Comparison of Isomaltulose and Sucrose"

_nutrients, 2021, doi:10.3390/nu13124493_

Round 1
Reviewer 1 Report
The following are my specific comments to your manuscript.
Lines 32-33: It is important to specify that it was brachial but not aortic stiffness.
Lines 43-44: In the study by Kelsch et al. (reference#9), high-glycemic index meal did not increase PWV. Thus, the statement is not supported by findings of this study.
Line 75: Were all 5 women postmenopausal?
Results:
Lines 208-209: include the P value of symbol shown in Figure 2B.
If baPWV incresead, but not hbPWV, these results indicate that arm but not aortic+leg arterial stiffness increased after SSI.
Lines 233-4: there is no HR of the ankle and DBP should be at the ankle or posterior tibial artery and not “of the ankle”. The ankle is a joint.
Lines 258-269: Fig 4 shows the correlation between baPWV (the lowest) and increases in blood glucose (the highest). The purpose of the study was to investigate the acute effects of isomaltose by increasing blood glucose on PWV. How was the correlation between baPWV and blood glucose at 90 min when there are the highest increase in baPWV?
Lines 271-273: the only time point at which both baPWV and SBP increased was at 90-min; however, blood glucose was not increased at this time point. Thus, the statement is inaccurate.
Lines 274-275: isomaltulose prevented the increase in baPWV, but did not reduce arterial stiffness.
Line 279, this statement is exaggerated since most of the day is > 12 hours.
Lines 287-288: Based on the findings of the present study, it is acceptable that isomaltulose may be able to slow down the rise in blood glucose observed with sucrose, but not compared with sucrose since there was no between group difference.
Lines 291-292: Please insert reference # 5 at the end of the findings of your previous study.
Lines 294-295: Use trial instead of trials since there was only one of each.
Line 299: PAD is specifically developed in the lower limbs or legs, but not in the arms.
Line 301-302: This statement is inaccurate because Tucker et al. (reference#25) measured brachial artery FMD and not peripheral arterial PWV. Insert a correct reference or change the statement.
Line 304: please provide a reference that showed that hbPWV is a measure of proximal aortic stiffness, which is measured by characteristic impedance (Zc).
Lines 314-315; The increased ADMA (an competitive inhibitor of eNOS) during hyperglycemia as a mechanism is for chronic hyperglycemia in T2D. Please provide a reference for acute hyperglycemia inducing endothelial dysfunction via increased ADMA.
Lines 319-324: Authors have to discuss how baPWV and SBP were increased at 90 post ingestion of Sucrose if blood glucose were not significantly elevated at this time point.
Line 337: add systemic before the second “arterial stiffness”. Without systemic, peripheral arterial stiffness may influence arterial stiffness reads redundant.
Lines 341-342: there is no evidence from your study that isomaltulose may help prevent arteriosclerosis, a pathological increase in collagen synthesis and elastin breakdown. Most likely the increase in baPWV after sucrose ingestion is not due to those structural changes but to impaired endothelial NO synthesis and increased sympathetic activation.
Your study did not show the ability of isomaltulose to reduce arterial stiffness. It prevented the increase in baPWV.
Author Response
Responses to Reviewer 1:
We appreciate your attention to our manuscript. We feel that our submission has been considerably enhanced by implementing your valuable advice. We addressed the specific comments as outlined in detail below, and modifications to the revised text are highlighted in red. We hope you will agree that the changes have rendered our report acceptable for publication.
This study conducted on a small group of subjects is interesting and quite original. However, this reviewer raises some issues that have to be addressed by authors.
The following are my specific comments to your manuscript.
Lines 32-33: It is important to specify that it was brachial but not aortic stiffness.
Response
We have corrected the text.
Lines 43-44: In the study by Kelsch et al. (reference#9), high-glycemic index meal did not increase PWV. Thus, the statement is not supported by findings of this study.
Response
We have changed the references.
Line 75: Were all 5 women postmenopausal?
Response
All women were postmenopausal.
Results:
Lines 208-209: include the P value of symbol shown in Figure 2B.
Response
We have corrected the text.
If baPWV incresead, but not hbPWV, these results indicate that arm but not aortic+leg arterial stiffness increased after SSI.
Response
We have made the changes as you thought.
Lines 233-4: there is no HR of the ankle and DBP should be at the ankle or posterior tibial artery and not “of the ankle”. The ankle is a joint.
Response
We have corrected the text.
Lines 258-269: Fig 4 shows the correlation between baPWV (the lowest) and increases in blood glucose (the highest). The purpose of the study was to investigate the acute effects of isomaltose by increasing blood glucose on PWV. How was the correlation between baPWV and blood glucose at 90 min when there are the highest increase in baPWV?
Response
There was no correlation between baPWV and blood glucose level after 90 minutes of intake.
This suggests that blood glucose levels are not directly involved in the increase in baPWV after ingestion.
In other words, sympathetic hyperactivity and increased oxidative stress associated with increased blood glucose levels may be the cause, so the text was changed.
Lines 271-273: the only time point at which both baPWV and SBP increased was at 90-min; however, blood glucose was not increased at this time point. Thus, the statement is inaccurate.
Response
We have corrected the text.
Lines 274-275: isomaltulose prevented the increase in baPWV, but did not reduce arterial stiffness.
Response
We have corrected the text.
Line 279, this statement is exaggerated since most of the day is > 12 hours.
Response
We have corrected the hyperbole.
Lines 287-288: Based on the findings of the present study, it is acceptable that isomaltulose may be able to slow down the rise in blood glucose observed with sucrose, but not compared with sucrose since there was no between group difference.
Response
Thank you for your understanding.
Lines 291-292: Please insert reference # 5 at the end of the findings of your previous study.
Response
We have added to the literature.
Lines 294-295: Use trial instead of trials since there was only one of each.
Response
We have corrected the text.
Line 299: PAD is specifically developed in the lower limbs or legs, but not in the arms.
Response
We have added an addendum.
Line 301-302: This statement is inaccurate because Tucker et al. (reference#25) measured brachial artery FMD and not peripheral arterial PWV. Insert a correct reference or change the statement.
Response
We have corrected the text.
Line 304: please provide a reference that showed that hbPWV is a measure of proximal aortic stiffness, which is measured by characteristic impedance (Zc).
Response
We added (Am J Hypertens. 2019 Jan 15;32(2):146-154).
Lines 314-315; The increased ADMA (an competitive inhibitor of eNOS) during hyperglycemia as a mechanism is for chronic hyperglycemia in T2D. Please provide a reference for acute hyperglycemia inducing endothelial dysfunction via increased ADMA.
Response
We have added an addendum.
Lines 319-324: Authors have to discuss how baPWV and SBP were increased at 90 post ingestion of Sucrose if blood glucose were not significantly elevated at this time point.
Response
We added.
We believe that sympathetic nervous system, vascular endothelial function, and oxidative stress are probably related in parallel.
Line 337: add systemic before the second “arterial stiffness”. Without systemic, peripheral arterial stiffness may influence arterial stiffness reads redundant.
Response
We have corrected the text.
Lines 341-342: there is no evidence from your study that isomaltulose may help prevent arteriosclerosis, a pathological increase in collagen synthesis and elastin breakdown. Most likely the increase in baPWV after sucrose ingestion is not due to those structural changes but to impaired endothelial NO synthesis and increased sympathetic activation.
Your study did not show the ability of isomaltulose to reduce arterial stiffness. It prevented the increase in baPWV.
Response
We have corrected the text.

Reviewer 2 Report
This is a very well-written study which investigates the effects of 2 carbohydrates with different glycemic index on arterial stiffness, glucose levels and BP at 30, 60 and 90 min after ingestion. Although the sample size is small, the results are interesting and may have important implications.
However, there are issues that should be addressed:
- SBP increased 90 min after sucrose ingestion. Authors suggested that the increase in SBP may be due to the increase in baPWV. However, the opposite (acute increase in PWV due to an acute increase in BP) may be a more prominent pathophysiological mechanism. Was there a relationship between SBP and baPWV at 90 min?
- baPWV and glucose levels were positively correlated 30 min after isomaltulose and sucrose ingestion. Given that there was no relationship between baPWV and glucose, how authors explain the increase in PWV 60 and 90 min after sucrose ingestion?
- The sample size is small. Did authors calculate the power of the study?
Author Response
Responses to Reviewer 2:
We appreciate your attention to our manuscript. We feel that our submission has been considerably enhanced by implementing your valuable advice. We addressed the specific comments as outlined in detail below, and modifications to the revised text are highlighted in red. We hope you will agree that the changes have rendered our report acceptable for publication.
This is a very well-written study which investigates the effects of 2 carbohydrates with different glycemic index on arterial stiffness, glucose levels and BP at 30, 60 and 90 min after ingestion. Although the sample size is small, the results are interesting and may have important implications.
However, there are issues that should be addressed:
- SBP increased 90 min after sucrose ingestion. Authors suggested that the increase in SBP may be due to the increase in baPWV. However, the opposite (acute increase in PWV due to an acute increase in BP) may be a more prominent pathophysiological mechanism. Was there a relationship between SBP and baPWV at 90 min?
Response
There was a relationship between SBP and baPWV at 90 minutes.
- baPWV and glucose levels were positively correlated 30 min after isomaltulose and sucrose ingestion. Given that there was no relationship between baPWV and glucose, how authors explain the increase in PWV 60 and 90 min after sucrose ingestion?
Response
We are considering the possibility that increased arterial stiffness during acute hyperglycemia may be related to increased sympathetic nerve activity and decreased vascular endothelial function associated with increased oxidative stress.
- The sample size is small. Did authors calculate the power of the study?
Response
Assessing the power of research. I wrote about it in Methods. We also wrote about the small sample size as a limitation of the study.

Round 2
Reviewer 1 Report
Discussion lines 3-4: Authors ignored this comment: isomaltulose prevented the increase in baPWV, but did not reduce arterial stiffness.
They say “we have corrected the text”, but they did not do it. The statement still says ”to reduce baPWV”, which is incorrect. It can be changed to “prevent the increase” in baPWV and SBP.
Paragraph before the conclusions: Adding “aortic and” to the sentence did not avoid the redundancy.
Change “arterial stiffness’ for baPWV. Aortic and peripheral arterial stiffness may influence baPWV.
In Conclusions: there is no evidence from your study that isomaltulose reduces the rapid rise in postprandial blood glucose levels. Isomaltulose “prevented” the rapid rise in postprandial blood glucose levels and the increased arterial stiffness and blood pressure.
Last two lines: Authors ignored my comments on the conclusions. Your study did not show that isomaltulose reduced arterial stiffness. ISI prevented the increase in baPWV. It is incorrect to recommend isomaltose to reduce arterial stiffness.
Author Response
Responses to Reviewer 1:
We appreciate your attention to our manuscript. We feel that our submission has been considerably enhanced by implementing your valuable advice. We addressed the specific comments as outlined in detail below, and modifications to the revised text are highlighted in red. We hope you will agree that the changes have rendered our report acceptable for publication.
Discussion lines 3-4: Authors ignored this comment: isomaltulose prevented the increase in baPWV, but did not reduce arterial stiffness.
They say “we have corrected the text”, but they did not do it. The statement still says ”to reduce baPWV”, which is incorrect. It can be changed to “prevent the increase” in baPWV and SBP.
Response: we have corrected the text.
Paragraph before the conclusions: Adding “aortic and” to the sentence did not avoid the redundancy.
Change “arterial stiffness’ for baPWV. Aortic and peripheral arterial stiffness may influence baPWV.
Response: we have corrected the text.
In Conclusions: there is no evidence from your study that isomaltulose reduces the rapid rise in postprandial blood glucose levels. Isomaltulose “prevented” the rapid rise in postprandial blood glucose levels and the increased arterial stiffness and blood pressure.
Response: we have corrected the text.
Last two lines: Authors ignored my comments on the conclusions. Your study did not show that isomaltulose reduced arterial stiffness. ISI prevented the increase in baPWV. It is incorrect to recommend isomaltose to reduce arterial stiffness.
Response: we have corrected the text.

Reviewer 2 Report
Authors have appropriately addressed the Reviewer's comments.
Minor changes should be made:
- Introduction, Abstract: Last two sentences should be rephrased: 'From the results of this study, it can be inferred that the results....' You better write: The results of the present study may have significant clinical implications on the implementation of dietary programs for middle-aged and elderly patients.
- Remove the sample size calculation to the Statistical Methods section or to an independent section.
- Discussion, line 3: These results suggest that isomaltulose could be used as an alternative to sucrose, given the neutral effect on PWV (omit the phrase 'to reduce' because it does not reflect the result of the study).
- Discussion, last 2 lines: Rephrase to 'In addition, although insulin and endothelial dysfunction may alter PWV, insulin levels and endothelial function biomarkers were not measured in the present study.
Author Response
Responses to Reviewer 2:
We appreciate your attention to our manuscript. We feel that our submission has been considerably enhanced by implementing your valuable advice. We addressed the specific comments as outlined in detail below, and modifications to the revised text are highlighted in red. We hope you will agree that the changes have rendered our report acceptable for publication.
Authors have appropriately addressed the Reviewer's comments.
Minor changes should be made:
- Introduction, Abstract: Last two sentences should be rephrased: 'From the results of this study, it can be inferred that the results....' You better write: The results of the present study may have significant clinical implications on the implementation of dietary programs for middle-aged and elderly patients.
Response: We have corrected the text.
- Remove the sample size calculation to the Statistical Methods section or to an independent section.
Response: We have corrected the text.
- Discussion, line 3: These results suggest that isomaltulose could be used as an alternative to sucrose, given the neutral effect on PWV (omit the phrase 'to reduce' because it does not reflect the result of the study).
Response: We have revised the text as advised.
- Discussion, last 2 lines: Rephrase to 'In addition, although insulin and endothelial dysfunction may alter PWV, insulin levels and endothelial function biomarkers were not measured in the present study.
Response: We have revised the text as advised.
